# FedSR: Frequency-Aware Enhancement for Diffusion-based Image Super-Resolution

## Abstract

Image super-resolution (ISR) is a classic and challenging problem in low-level vision because the data collection process often introduces complex and unknown degradation patterns. Leveraging powerful generative priors, diffusion-based algorithms have recently established new state-of-the-art ISR performance. Despite the promise, current diffusion-based ISR methods mostly focus on the spatial domain. To bridge this gap, we first experimentally validate that the key to solving the ISR problem lies in addressing the degradation of image amplitude information and high-frequency details. Based on this, we propose a novel *training-free* frequency-aware enhancement framework (**FedSR**) for diffusion-based ISR methods, which consists of two critical components. Firstly, we design the Amplitude Enhancement Module (AEM), which selectively enhances crucial amplitude channels through weighted optimization. Secondly, we introduce the High-Frequency Enhancement Module (HEM) that adaptively masks the skip features to perform high-pass filtering. Through extensive evaluations on both synthetic datasets and real-world image collections, our method demonstrates outstanding performance in reproducing realistic image details without additional tuning. For instance, FedSR improves StableSR across three datasets by **+10.53**% on MUSIQ metric.

## 1 Introduction

Image super-resolution (ISR) is a fundamental task in low-level vision that aims to reconstruct high-resolution (HR) images from their low-resolution (LR) counterparts. It has widespread applications in areas such as medical imaging (Li et al., 2024; Mao et al., 2023b), satellite imagery (Shermeyer & Etten, 2019; Cornebise et al., 2022), and surveillance systems (Liu et al., 2017; Liang, 2021), where obtaining high-quality images can naturally be subject to hardware limitations and transmission losses. Early ISR algorithms (Dong et al., 2016a; Tai et al., 2017; Chen et al., 2021) attempt to construct synthetic image pairs through simple handcrafted degradation operations (e.g., bicubic downsampling). However, they fail to generalize well in realistic scenarios since real-world LR images typically involve more complex and unknown degradation patterns.

To address this problem, some work (Zhang et al., 2021; Wang et al., 2021) resorts to Generative Adversarial Networks (GAN) (Goodfellow et al., 2014) to enhance visual perception generated by using the adversarial training loss. However, these methods tend to introduce unpleasant visual artifacts because of the instability of adversarial training. Recently, a series of studies (Wang et al., 2023c; Lin et al., 2023; Yu et al., 2024; Wu et al., 2023; Yang et al., 2023) have discovered that incorporating diffusion priors (Rombach et al., 2022) can result in realistic restoration results, achieving state-of-the-art (SOTA) ISR performance. For example, StableSR (Wang et al., 2023c) trains a time-aware encoder to guide Stable Diffusion (Rombach et al., 2022) to achieve promising restoration results; DiffBIR (Lin et al., 2023) employs an IRControlNet trained based on condition images to generate realistic details. Despite the promising results, current diffusion-based ISR methods operate solely in the spatial domain and lack a deep understanding of the frequency domain.

To explore the opportunity to improve diffusion-based ISR models from a frequency perspective, we refer to the following well-established observations: **(1) Loss of High-Frequency Details**: Image degradation often leads to the loss of high-frequency details. **(2) Degradation of Amplitude:** Inspired by tasks such as dehazing (Yu et al., 2022) and deraining (Guo et al., 2022), image degradation can also result in the loss of amplitude information. To systematically validate these phenomena

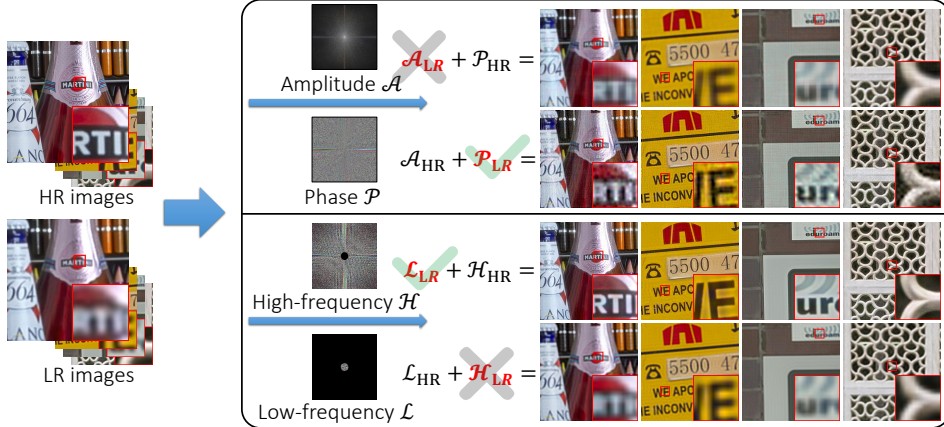

Figure 1: The impact of the real-world degradation on each component. **Top:** We replace the amplitude and phase components of the original HR image ($\mathcal{A}_{HR}$ and $\mathcal{P}_{HR}$) with the corresponding components from the degraded LR image ($\mathcal{A}_{LR}$ and $\mathcal{P}_{LR}$). **Bottom:** Similarly, the original low- and high-frequency components $\mathcal{L}_{HR}$ and $\mathcal{H}_{HR}$ are replaced with $\mathcal{L}_{LR}$ and $\mathcal{H}_{LR}$.

and facilitate readers' understanding, we conducted additional experiments (see Figure 1 and Appendix B). From a technical perspective, researchers have explored various frequency-based ISR algorithms. However, early efforts focus on improving traditional model architectures like ResNet (He et al., 2016) and GANs (Goodfellow et al., 2014). Though several recent works (Luo et al., 2023; Wang et al., 2024b; Zhao et al., 2024; Moser et al., 2024) also explore improving diffusion-based ISR, they rely on heavy training processes and handcrafted network structure modifications.

In this paper, we propose a generic and *training-free* **F**requency-aware **E**nhancement framework for **D**iffusion-based **S**uper-**R**esolution (dubbed **FedSR**). Specifically, FedSR encapsulates two key components. **(1) Amplitude Enhancement Module**: To enhance the lost amplitude components, we develop an amplitude enhancement module (AEM) that utilizes a channel-aware mechanism to enhance the amplitude components which convey crucial details. **(2) High-Frequency Enhancement Module:** We further design a high-frequency enhancement module (HEM) that operates on the skip connection features, which employs a spectral modulation method to adaptively enhance the prominent high-frequency information in the skip features. The two modules can be simultaneously integrated into current diffusion-based ISR models, without requiring any further fine-tuning. Through extensive experiments, our FedSR significantly improves state-of-the-art diffusion-based ISR algorithms StableSR, PASD by **+10.53**%, **+10.67**% on MUSIQ metric, respectively. These results clearly validate the superiority of our FedSR algorithm in enhancing the amplitude and high-frequency details from a frequency perspective.

The main contributions of our work are as follows: **(A General framework)** We present a **general framework** that is able to improve most diffusion-based SR algorithms without extra training costs. **(Technical Novelty)** Motivated by our empirical findings, we propose a novel channel selection mechanism for enhancing the amplitude information. Also, we develop a new semantic-aware high-pass filtering algorithm that adaptively determines the thresholds by feature inputs. Again, we note that the two modules are **totally training-free. (Experiments)** We conduct extensive experiments on three benchmarks, demonstrating that FedSR improves 5 SOTA diffusion-based SR methods, verifying its generality. Moreover, we have no extra training cost, maintaining almost the same complexity parameters.

## 2 RELATED WORK

**Image Super-Resolution (ISR).** Although deep learning-based ISR techniques have gained widespread adoption, most CNN-based methods (Dong et al., 2016a; Lim et al., 2017; Kim et al., 2016; Dong et al., 2016b; Shi et al., 2016) still suffer from the issue of excessive detail smoothing. To address this problem and better enhance visual perception, recent advances (Zhang et al., 2021;

Wang et al., 2021; Liang et al., 2021; Chen et al., 2022; Liang et al., 2022; Wang et al., 2024a) using the GAN-based models in the field of Real-ISR have explored more complex degradation models for adversarial training. For instance, BSRGAN (Zhang et al., 2021) synthesizes more realistic degradation by using a random shuffling strategy, and RealESRGAN (Wang et al., 2021) employs high-order degradation modeling techniques. While these methods have made progress in generating more perceptually realistic details, GAN-based ISR methods often suffer from unstable adversarial training, frequently introducing unnatural visual artifacts. In recent years, the powerful Stable Diffusion (SD) (Rombach et al., 2022) model has been applied to ISR tasks (Wang et al., 2023c; Lin et al., 2023; Yu et al., 2024; Wu et al., 2023; Yang et al., 2023; Wang et al., 2023d; Cui et al., 2024). For instance, PASD (Yang et al., 2023) utilizes pixel-aware cross attention to perceive image local structures. SUPIR (Yu et al., 2024) develops a trimmed ControlNet (Zhang et al., 2023) and ZeroSFT to reduce the model size. Although these methods demonstrate excellent performance in real-world ISR tasks, they are limited to operations in the spatial domain and do not thoroughly explore the characteristics of the frequency domain. In contrast, we discuss the degradation processes of various frequency components and design a training-free method to enhance these degraded components.

**Frequency-based Super-Resolution.** Frequency analysis of image processing has been widely used in computer vision (Yu et al., 2022; Huang et al., 2024; Yang & Soatto, 2020; Cai et al., 2021; Si et al., 2023; Yu et al., 2022; Ji et al., 2021). For super-resolution tasks, many studies improve images reconstruction quality by applying frequency domain transformations to comprehensively extract feature information from low-resolution images (Guan et al., 2024; Li et al., 2023a; Xu et al., 2024; Xie et al., 2021). Some methods enhance performance by constructing frequency domain loss functions that focus on recovering frequency information through heavy network training (Zhu et al., 2023; Fuoli et al., 2021; Dong et al., 2023; Ji et al., 2021; Wang et al., 2024c; Li et al., 2023b). For example, Fuoli et al. (2021) designs Fourier space supervision losses to enhance perceptual quality in image super-resolution. Additionally, some methods improve reconstruction quality by separating specific components (such as high-frequency components) in the frequency domain (Guan et al., 2024; Li et al., 2023a; Xu et al., 2024; Xie et al., 2021; Dai et al., 2024; Yang et al., 2022a; Jiang et al., 2023). Appendix A.1 lists the effects of different frequency components on image quality for other computer vision tasks. Although these existing frequency domain-based ISR methods significantly improve performance, they have two main drawbacks: first, they rely on frequency domain loss functions to achieve realistic outcomes with heavy training; second, they typically focus only on certain specific components in the frequency domain. In contrast, our training-free FedSR systematically analyzes the degradation process from the perspective of image modeling and then enhances these degraded components.

## 3 BACKGROUND AND PRELIMINARIES

### 3.1 DIFFUSION MODELS FOR IMAGE SUPER-RESOLUTION

Diffusion models, such as DDPM (Ho et al., 2020) and LDM (Rombach et al., 2022), are a class of latent variable models, which primarily consist of a diffusion process and a denoising process. In the diffusion process, Gaussian noise is gradually added at each time step $t$ according to a predefined variance schedule denoted as $\beta_1, ..., \beta_t$, via a Markov chain. It eventually results in a random noise distribution, which is defined as,

$$q\left(\boldsymbol{x}_t \mid \boldsymbol{x}_{t-1}\right) = \mathcal{N}\left(\boldsymbol{x}_t; \sqrt{1 - \beta_t}\boldsymbol{x}_{t-1}, \beta_t \mathcal{I}\right). \tag{1}$$

In the denoising process, given the noisy input $\boldsymbol{x}_t$, the model outputs the clean data $\boldsymbol{x}_{t-1}$ before noise is added, which is represented as,

$$p_\theta\left(\boldsymbol{x}_{t-1} \mid \boldsymbol{x}_t\right) = \mathcal{N}\left(\boldsymbol{x}_{t-1}; \boldsymbol{\mu}_\theta\left(\boldsymbol{x}_t, t\right), \boldsymbol{\Sigma}_\theta\left(\boldsymbol{x}_t, t\right)\right). \tag{2}$$

Here, $\boldsymbol{\mu}_\theta$ and $\boldsymbol{\Sigma}_\theta$ are determined by the denoising model. Current diffusion-based generative models (Ho et al., 2020; Rombach et al., 2022) are implemented using a U-Net (Ronneberger et al., 2015) architecture to remove noise from data samples, which consists of a contracting path for downsampling and an expansive path for upsampling. Each upsampling block concats both the backbone and skip features in the skip connections.

To ensure that diffusion-based generative models meet the requirements for image quality and fidelity in ISR tasks, existing methods typically utilize LR images to guide model generation. First,

the LR image is used as a conditional input and transformed into an embedding through the image encoder. Then, these embeddings are fused with the U-Net using a cross-attention mechanism or a custom control module to guide the generation of HR images. Through iterative diffusion and reverse processes, these models effectively capture complex image features, enhancing the capability to recover realistic details.

## 3.2 FOURIER FREQUENCY DOMAIN TRANSFORMATION

The Fast Fourier Transform (FFT) is widely applied in low-level vision tasks, transforming images from the spatial domain to the Fourier domain, denoted as,

$$\mathcal{F}(\boldsymbol{x})(u,v) = \sum_{h=0}^{H-1}\sum_{w=0}^{W-1}\boldsymbol{x}(h,w)e^{-j2\pi\left(\frac{h}{H}u+\frac{w}{W}v\right)}. \tag{3}$$

Its inverse function (IFFT) is formulated as,

$$\mathcal{G}(\boldsymbol{f})(h,w) = \frac{1}{UV}\cdot\sum_{u=0}^{U-1}\sum_{v=0}^{V-1}\boldsymbol{f}(u,v)e^{-j2\pi\left(\frac{u}{U}h+\frac{v}{V}w\right)}, \tag{4}$$

where $j$ is the imaginary unit; $e$ is Euler's number, which can be formulated as $e^{j\theta} = \cos\theta + j\sin\theta$. $\mathcal{F}(\cdot)$ and $\mathcal{G}(\cdot)$ are 2D Fourier transform and inverse 2D Fourier transform, respectively. The frequency features $\mathcal{F}(\boldsymbol{x})$ in Eq. (3) and $\boldsymbol{f}$ in Eq. (4) are both tensors in complex domain, expressed as $\mathcal{F}(\boldsymbol{x}) = \mathcal{R}(\boldsymbol{x})+j\mathcal{I}(\boldsymbol{x})$, where $\mathcal{R}(\boldsymbol{x})$ and $\mathcal{I}(\boldsymbol{x})$ are the real parts and imaginary parts, respectively.

In this paper, we explore two decomposition methods in the frequency domain, and the related analysis refers to Appendix A. The first is composition-based decomposition, which separates frequency into the amplitude $\mathcal{A}$ and phase $\mathcal{P}$, represented as,

$$\mathcal{A}(\boldsymbol{x})(u,v) = \sqrt{\mathcal{R}^2(\boldsymbol{x})(u,v) + \mathcal{I}^2(\boldsymbol{x})(u,v)},$$
$$\mathcal{P}(\boldsymbol{x})(u,v) = \arctan\left[\frac{\mathcal{I}(\boldsymbol{x})(u,v)}{\mathcal{R}(\boldsymbol{x})(u,v)}\right]. \tag{5}$$

The other method is distance-based decomposition, where we divide the frequency information into high-frequency and low-frequency parts based on their distance from the frequency center.

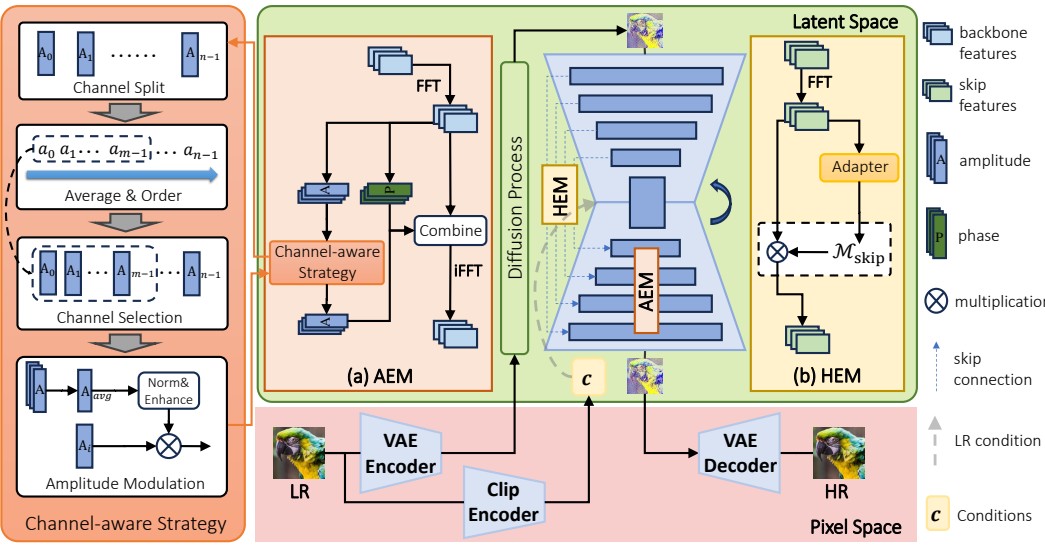

Figure 2: The overview of FedSR, which has two modules, (a) AEM: a channel-aware amplitude enhancement module which selectively enhances crucial amplitude channels through reweighting strategy; (b) HEM: a high-frequency enhancement module which utilizes adaptive masking.

# 4 THE PROPOSED FRAMEWORK

In this section, we describe our novel FedSR framework in detail. Essentially, FedSR comprises two key components: an amplitude modulation module that operates on the backbone features (Section 4.1), and a spectral modulation module designed for adaptive enhancement of high-frequency components (Section 4.2). Importantly, both modules are post-hoc adjustments to diffusion models, requiring no additional heavy tuning. Notably, FedSR can be seamlessly integrated as plugins into any off-the-shelf diffusion-based ISR models. The overall architecture is shown in Figure 2.

## 4.1 CHANNEL-AWARE AMPLITUDE ENHANCEMENT

In our preliminary experiments, we demonstrate the phenomenon of amplitude degradation in LR images. Thus, during the training process, the ISR model would actively learn the amplitude signal features from HR images. However, due to the black-box nature of DNNs, enhancing amplitude features cannot be directly achieved by simply following traditional image processing conclusions and requires further exploration; see Appendix D.1 for further discussion.

**Analysis of Amplitude Channels.**     Inspired by some studies (Hu et al., 2018; Zhao et al., 2019), which enhance model performance by adjusting the importance of different channel features in convolutional neural networks (CNNs), we hypothesize that the amplitude features of various channels in the U-Net backbone network which contains convolutional layers, may also convey information of varying significance. To validate this hypothesis, we transform the image features generated by Stable Diffusion (Rombach et al., 2022) into the frequency domain and then select the channels according to their amplitude values. Figure 3 (a) illustrates the reconstructed images using different channels at different sampling steps. Our observations reveal that amplitude channels with lower amplitude values convey crucial details of the image, while channels with higher amplitude values result in disorganized and chaotic images, indicating that these channels have learned meaningless signals. This underscores the importance of emphasizing significant amplitude features during the sampling of ISR models to enhance image quality.

Based on this finding, we develop a simple yet effective channel-aware Amplitude Enhancement Module (AEM) aimed at selectively modulating the amplitude information in the backbone network by identifying channels with rich information, thereby improving the overall visual quality of the images. Technically, the AEM first transforms the U-Net backbone features before the concatenation of skip connections in upsampling blocks into the frequency domain. Then it extracts the amplitude components with Eq. (5) as the optimization target. Subsequently, we design the aforementioned channel-aware strategy, which consists of the following four steps.

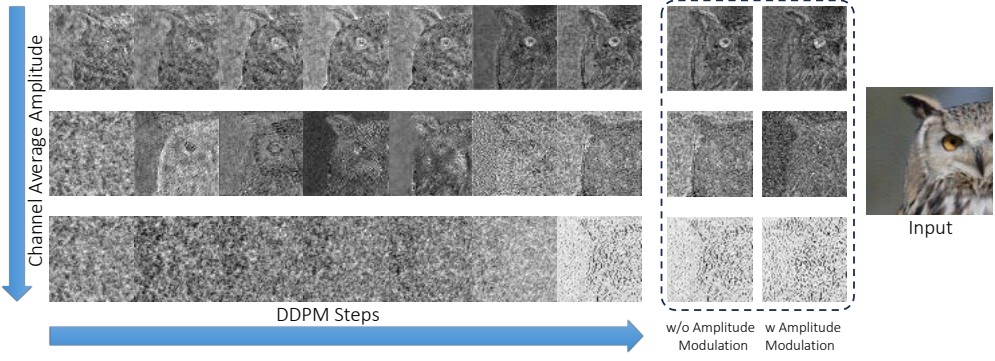

(a) Channel-wise Visualization                              (b) Effects of Amplitude Modulation

Figure 3: Average amplitude on features and effect of amplitude modulation. (a) During the generation process of the diffusion model, lower average amplitude in the channel leads to clearer generated images. (b) The application of our amplitude modulation further enhances feature clarity.

**a) Channel Separation.** Inspired by SENet's (Hu et al., 2018) different processing of features across varying channels, we split the amplitude component along the channel dimension and then obtain the amplitude set with all channels, denoted as $\boldsymbol{\mathcal{S}}_A = \{\mathcal{A}(\boldsymbol{x}_{\text{bone}})_i\}_{i=1}^{C}$, to separate various pieces of information, where $\mathbf{x}_{\text{bone}}$ is the backbone features; $C$ is the number of amplitude channels.

**b) Average Amplitude Value Ranking.** Recall that channels with lower average amplitude generally exhibit clearer details in Figure 3 (a). Therefore, we compute the average value of the amplitude component for each channel, formulated as $a_i = \frac{1}{HW} \sum_{h=1}^{H} \sum_{w=1}^{W} \mathcal{A}(\boldsymbol{x}_{\text{bone}})_i^{(h,w)}$, where $\mathcal{A}(\boldsymbol{x}_{\text{bone}})_i = (\mathcal{A}(\boldsymbol{x}_{\text{bone}})_i^{(h,w)})_{H \times W}$ with $H$ as the height of the feature and $W$ as the width of the feature. Then we rank the amplitude of each channel $\mathcal{A}(\boldsymbol{x}_{\text{bone}})_i$ in $\boldsymbol{\mathcal{S}}_A$ in ascending order based on the average value $a_i$, to identify channels with richer detailed information.

**c) Channel Selection.** Based on the ranking results, we select channels with lower amplitude values that contain abundant information and then combine them into a subset $\boldsymbol{\mathcal{S}}_S = \{\mathcal{A}(\boldsymbol{x}_{\text{bone}})_i | a_i \leq a_{\min} + P_s \times (a_{\max} - a_{\min})\}$ for subsequent amplitude modulation. Here, $P_s$ is the selection thresholds. To achieve better results of AEM, we design a supplementary experiment exhaustively testing $P_s$ between 0 and 1 on the validation set, shown in Section 5.3.

**d) Amplitude Modulation.** To amplify the impact of these selected amplitude channels, we apply amplitude reweight at the final step of sampling. Specifically, to align the amplitude components with their size characteristics, we first compute the average amplitude $\overline{\boldsymbol{\mathcal{A}}} = \frac{1}{C} \sum_{i=1}^{C} \mathcal{A}(\boldsymbol{x}_{\text{bone}})_i$ along the channels, followed by linear normalization to construct a factor map

$$\boldsymbol{\mathcal{M}}_{\text{bone}} = 1 - P_a \cdot \frac{\overline{\boldsymbol{\mathcal{A}}} - \overline{\mathcal{A}}_{\min}}{\overline{\mathcal{A}}_{\max} - \overline{\mathcal{A}}_{\min}}, \tag{6}$$

where $\overline{\mathcal{A}}_{\min}$ and $\overline{\mathcal{A}}_{\max}$ means the minimum and maximum of $\overline{\boldsymbol{\mathcal{A}}}$; $P_a$ is a positive linearization parameter. We then multiply the factor map $\boldsymbol{\mathcal{M}}_{\text{bone}}$ with the channel features in the subset $\mathcal{S}_S$ one by one, formulated as follows,

$$\mathcal{A}(\boldsymbol{x}_{\text{bone}})_i' = \left\{ \begin{array}{ll} \mathcal{A}(\boldsymbol{x}_{\text{bone}})_i \odot \boldsymbol{\mathcal{M}}_{\text{bone}}, & \text{if } i \in \boldsymbol{\mathcal{S}}_S; \\ \mathcal{A}(\boldsymbol{x}_{\text{bone}})_i, & \text{otherwise.} \end{array} \right. \tag{7}$$

To apply the enhanced amplitude, we use the modulated amplitude and the original phase components to combine into the frequency domain by $\mathcal{F}(\boldsymbol{x}_{\text{bone}})' = \mathcal{R}(\boldsymbol{x}_{\text{bone}})' + j\mathcal{I}(\boldsymbol{x}_{\text{bone}})'$, and further transfer to the spatial domain by the inverse Fourier transformation $\boldsymbol{x}_{\text{bone}}' = \mathcal{G}(\mathcal{F}(\boldsymbol{x}_{\text{bone}})')$.

There is a subtle point worth deeper discussion. At first glance, it might seem counterintuitive that reducing the amplitude values in Eq. (6) and Eq. (7) would enhance image super-resolution (ISR). However, our further experiments (see Appendix D.1) indicate that the logic behind traditional image processing may differ from that of diffusion networks. Increasing the amplitude of the original image signal typically affects image contrast and brightness. In contrast, within FedSR, reducing the amplitude of the deep feature signals results in clearer detail. We speculate the reason might be that diffusion models are prone to highlight channels with smaller amplitude values. Further exploration of this behavior requires deeper theoretical insights from diffusion models in the frequency domain, which we leave for future work.

### 4.2 Adaptive Masking for High-frequency Enhancement

Next, we discuss our modification to diffusion models to enhance ISR performance from the perspective of high-frequency details. Inspired by FreeU (Si et al., 2023), we know that the skip connections in U-Net blocks can transmit high-frequency, information-rich features to deeper layers of the network, thereby preserving more comprehensive image information. Note that FreeU is designed for text-to-image tasks which only applies two constant scaling transformations to low-frequency features on all layers to achieve high-pass filtering. However, for the diffusion-based ISR problems, the features on different U-Net layers convey various semantic information. Therefore, considering the varying richness of information, we propose a high-frequency enhancement module (HEM) with adaptive masking, which can be divided into the following two steps; see Figure 2 (b).

Table 1: Quantitative comparison with SOTA methods on the synthetic benchmark DIV2K-Val (Agustsson & Timofte, 2017). **Bold** and $\Delta$ represent the improvement and the performance boost brought by FedSR, respectively. Red and blue colors represent the best and second-best performance. ↓ represents the smaller the better, while ↑ represents the opposite.

| Method | PSNR↑ | SSIM↑ | LPIPS↓ | CLIP-IQA↑ | MUSIQ↑ | NIQE↓ | MANIQA↑ |
|---|---|---|---|---|---|---|---|
| **StableSR** (IJCV2024) | 23.26 | 0.5644 | 0.3119 | 0.6771 | 65.91 | 4.742 | 0.4208 |
| **FedSR+StableSR** | 22.59 | 0.553 | 0.3711 | 0.7275 | 71.48 | **4.112** | 0.4914 |
| **△StableSR** | -0.67 | -0.0114 | +0.0592 | **+0.0504** | **+5.57** | **-0.63** | **+0.0706** |
| **SUPIR** (CVPR2024) | 22.14 | 0.5180 | 0.3930 | 0.7130 | 63.60 | 5.705 | 0.5533 |
| **FedSR+SUPIR** | 20.98 | 0.4847 | 0.4125 | 0.7368 | 66.57 | 5.437 | **0.5841** |
| **△SUPIR** | -1.16 | -0.0333 | +0.0195 | **+0.0238** | **+2.97** | **-0.268** | **+0.0308** |
| **SeeSR** (CVPR2024) | 23.67 | 0.5978 | 0.3200 | 0.6940 | 68.72 | 4.806 | 0.5044 |
| **FedSR+SeeSR** | 23.56 | 0.6101 | 0.3401 | 0.6893 | 70.21 | 4.598 | 0.5184 |
| **△SeeSR** | -0.11 | +0.0123 | +0.0201 | -0.0047 | **+1.49** | **-0.208** | **+0.0140** |
| **PASD** (ECCV2024) | 24.16 | 0.6099 | 0.3705 | 0.5848 | 61.85 | 5.169 | 0.4028 |
| **FedSR+PASD** | 24.25 | 0.6213 | 0.3644 | 0.5948 | 65.72 | 4.904 | 0.4223 |
| **△PASD** | +0.09 | +0.0114 | -0.0061 | **+0.0100** | **+3.87** | **-0.265** | **+0.0195** |
| **DiffBIR** (Arxiv2023) | 23.14 | 0.5370 | 0.3667 | 0.7301 | 69.90 | 4.991 | 0.5675 |
| **FedSR+DiffBIR** | 22.38 | 0.5222 | 0.4236 | **0.7382** | **73.04** | 4.729 | 0.5838 |
| **△DiffBIR** | -0.76 | -0.0148 | +0.0569 | **+0.0081** | **+3.14** | **-0.262** | **+0.0163** |

**a) Adaptive Mask Construction.** To accurately filter and dynamically enhance the high-frequency components in the skip features, we construct an adaptive high-frequency mask $\mathcal{M}_{\text{skip}}$. Considering that lower-level and smaller-scale features often contain less image detailed information, the mask adjusts the enhancement factor based on scale adaptively, to better adapt to the frequency structure of features at different levels, formulated as,

$$\mathcal{M}_{\text{skip}}(r) = \begin{cases} 1 + \left( \frac{S - S_{\min}}{S_{\max} - S_{\min}} + 0.5 \right) \cdot \frac{P_b}{2}, & \text{if } r > r_{\text{thresh}}; \\ 1, & \text{otherwise.} \end{cases} \tag{8}$$

Here $S$ is the scale of skip features, and $P_b$ is the enhancement factor; $r$ and $r_{\text{thresh}}$ are the radius and the radius threshold, respectively. Thus, the high-frequency components are split through masking.

**b) High-Frequency Component Enhancement.** We then multiply the adaptive mask $\mathcal{M}_{\text{skip}}$ element-wise with the skip features $\boldsymbol{x}_{\text{skip}}$ in the frequency domain to amplify and enhance the high-frequency components, represented as,

$$\mathcal{F}(\boldsymbol{x}_{\text{skip}})' = \mathcal{F}(\boldsymbol{x}_{\text{skip}}) \odot \mathcal{M}_{\text{skip}}, \tag{9}$$

where $\odot$ denotes element-wise multiplication. Finally, the inverse Fourier transformation, which is denoted as $\boldsymbol{x}'_{\text{skip}} = \mathcal{G}(\mathcal{F}(\boldsymbol{x}_{\text{skip}})')$, transfers the enhanced skip features to the spatial feature domain.

**Remark.** In practical applications, the AEM and HEM modules can actually be integrated into any layer of the diffusion U-Net blocks. However, our experimental validation shows that the better setup is to apply the AEM to the backbone features and the HEM to the skip features, as this configuration consistently yields superior performance; we may refer the readers to Appendix D.2 for more discussion. Empirically, both modules can be simultaneously incorporated into diffusion-based ISR models without the need for additional fine-tuning or adjustments. On various ISR benchmarks, FedSR achieves significant performance gains, effectively offering a *free lunch* for ISR.

## 5 Experiments

### 5.1 Experimental Settings

**Datasets and Baselines.** We employ the test datasets from StableSR (Wang et al., 2023c) and evaluate our approach on both synthetic and real-world datasets. (1) For the synthetic dataset, we

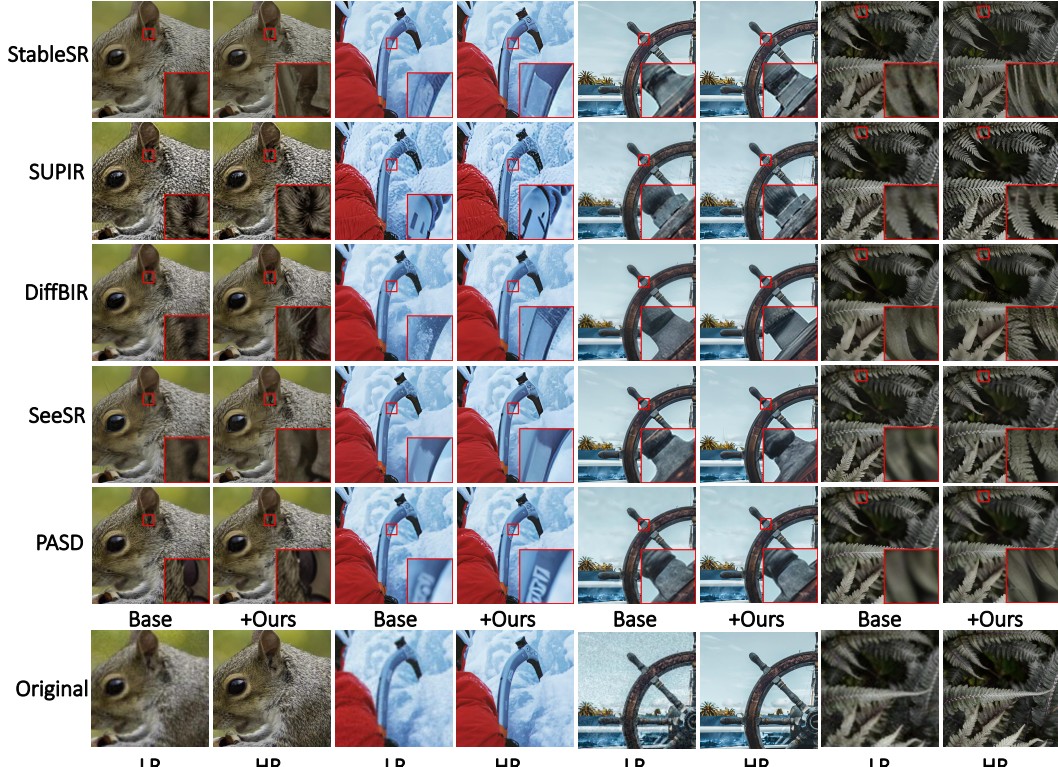

Figure 4: Qualitative comparisons of diffusion model-based ISR methods before and after incorporating our FedSR. It shows that FedSR can reconstruct more realistic HR images.

use 3,000 generated pairs of LR-HR images from the DIV2K validation set (Agustsson & Timofte, 2017), where the LR images have a resolution of $128 \times 128$, and the HR images have a resolution of $512 \times 512$. (2) For the real-world datasets, we utilize the DRealSR (Wei et al., 2020) and RealSR (Cai et al., 2019) datasets center-cropping the LR images to $128 \times 128$.

We select five state-of-the-art (SOTA) diffusion-based ISR models, namely StableSR (Wang et al., 2023c), SUPIR (Yu et al., 2024), SeeSR (Wu et al., 2023), PASD (Yang et al., 2023), and DiffBIR (Lin et al., 2023). And we incorporate our FedSR into these frameworks to evaluate effectiveness.

**Evaluation Metrics.**     We adopt a series of full-reference and no-reference metrics to assess the performance of different methods. The full-reference metrics includes PSNR, SSIM (evaluated on the Y channel in the YCbCr color space), and LPIPS (Zhang et al., 2018). For quality evaluation, we employ no-reference image quality assessment (IQA) metrics: CLIP-IQA (Wang et al., 2023b), MUSIQ (Ke et al., 2021), NIQE (Zhang et al., 2015), and MANIQA (Yang et al., 2022b).

**Implementation Details.**     To obtain the validation set of LR-HR pairs, we employ the degradation process of BSRGAN (Zhang et al., 2021) on the small random subset of size 100 from the DIV2K training set. Then we adjust the hyper-parameters in FedSR. Based on the default settings (i.e., $P_a = 0.5$, $P_{b1} = 1$, and $P_{b2} = 1$) as default, for further tuning. To determine the selection threshold $P_s$, we experiment with $P_s \in [0, 1]$ (see Figure 6) on a validation set created also by randomly selecting from DIV2K. We find that as the channel selection threshold increases, various metrics gradually stabilize, and set $P_s$ as 0.3 for better performance, further indicating larger amplitude channels contribute less. Detailed hyper-parameter settings can be found in the Appendix D.

## 5.2 COMPARISON BEFORE AND AFTER FEDSR APPLICATION

**Quantitative Comparisons.**     As shown in Table 1, we apply our method to five SOTA diffusion-based ISR frameworks, and the results on DIV2K-Valid indicate that almost all no-reference metrics

Table 2: Quantitative results on the real-world benchmark DRealSR with our FedSR.

| Method | PSNR↑ | SSIM↑ | LPIPS↓ | CLIP-IQA↑ | MUSIQ↑ | NIQE↓ | MANIQA↑ |
|---|---|---|---|---|---|---|---|
| **StableSR** (IJCV2024) | 27.93 | 0.7442 | 0.3280 | 0.6272 | 58.28 | 6.475 | 0.3890 |
| **FedSR+StableSR** | 26.68 | 0.7206 | 0.3903 | 0.6690 | 67.27 | **5.373** | 0.4810 |
| **△StableSR** | -1.25 | -0.0236 | +0.0623 | **+0.0418** | **+8.99** | **-1.102** | **+0.0920** |
| **SUPIR** (CVPR2024) | 24.80 | 0.6333 | 0.4323 | 0.6880 | 59.73 | 7.420 | 0.5040 |
| **FedSR+SUPIR** | 23.18 | 0.5777 | 0.4622 | **0.7232** | 64.09 | 6.810 | 0.5584 |
| **△SUPIR** | -1.62 | -0.0556 | +0.0299 | **+0.0352** | **+4.36** | **-0.610** | **+0.0544** |
| **SeeSR** (CVPR2024) | 28.04 | 0.7661 | 0.3188 | 0.6924 | 65.08 | 6.389 | 0.5134 |
| **FedSR+SeeSR** | 27.33 | 0.7671 | 0.3422 | 0.6944 | 67.46 | 6.052 | 0.5300 |
| **△SeeSR** | -0.71 | +0.0010 | +0.0234 | **+0.0020** | **+2.38** | **-0.337** | **+0.0166** |
| **PASD** (ECCV2024) | 28.96 | 0.7919 | 0.3142 | 0.5122 | 52.29 | 6.929 | 0.3672 |
| **FedSR+PASD** | 28.28 | 0.7860 | 0.3203 | 0.5790 | 62.16 | 6.420 | 0.4232 |
| **△PASD** | -0.68 | -0.0059 | +0.0061 | **+0.0668** | **+9.87** | **-0.509** | **+0.0560** |
| **DiffBIR** (Arxiv2023) | 25.90 | 0.6220 | 0.4715 | 0.7076 | 66.22 | 6.309 | 0.5568 |
| **FedSR+DiffBIR** | 24.53 | 0.6014 | 0.5024 | 0.7167 | **71.90** | 5.833 | **0.5902** |
| **△DiffBIR** | -1.37 | -0.0206 | +0.0309 | **+0.0091** | **+5.68** | **-0.476** | **+0.0334** |

improved. It suggests that our FedSR can further enhance image quality within these existing frameworks. Table 2 and Table 3 present the results on real-world datasets. For example, on the DIV2K-Val dataset, FedSR improves the original StableSR by **+7.44**% on CLIP-IQA metric. Additionally, on the real-world datasets DRealSR and RealSR, our method improves PASD by **+18.88**% and **+6.88**% on MUSIQ metric, respectively, thus demonstrating the effectiveness of FedSR. Although our method does not show significant improvements in full-reference metrics (PSNR, SSIM, and LPIPS), these metrics only capture certain aspects of performance (Blau & Michaeli, 2018; Ledig et al., 2017). Moreover, Figure 5 shows that solely pursuing improvements in these traditional metrics does not necessarily lead to better visual effects. Our FedSR, while maintaining reasonable PSNR/SSIM, significantly enhances no-reference metrics (largely improved MUSIQ +**10.53**%).

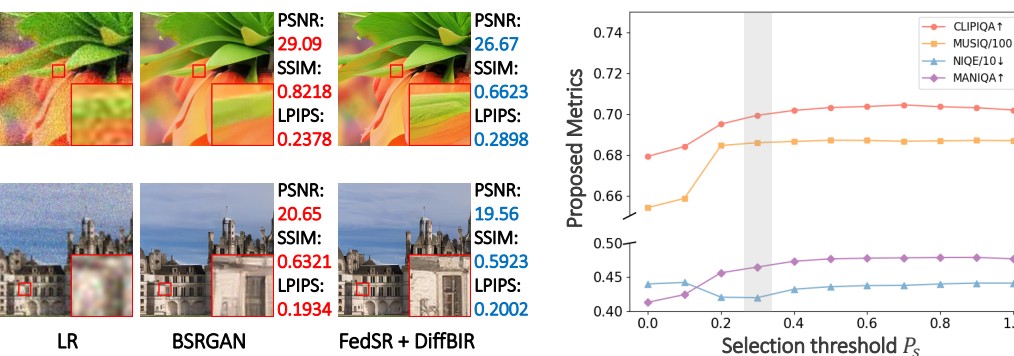

Figure 5: Ours (FedSR+DiffBIR) generates images with better image quality but obtains lower metrics in PSNR, SSIM, and LPIPS, which shows the bias between metric evaluation and image quality.

Figure 6: Selection thresholds and performance on our validation set (a random subset DIV2K training data). The gray column represents the best selection threshold $P_s$.

**Qualitative Comparisons.** To demonstrate the effectiveness of FedSR, Figure 4 presents a comparison before and after incorporating FedSR. It can be observed that our method significantly enhances the quality of the image generated by diffusion-based ISR methods, particularly in detailed textures and general visual effects. An interesting observation is that there occurs pseudo-textures in some images (e.g., squirrels) when applying FedSR to baselines like SeeSR. However, a closer inspection shows that the original baselines already demonstrate pseudo textures (though

Table 3: Quantitative results on the real-world benchmark RealSR with our FedSR.

| Method | PSNR↑ | SSIM↑ | LPIPS↓ | CLIP-IQA↑ | MUSIQ↑ | NIQE↓ | MANIQA↑ |
|---|---|---|---|---|---|---|---|
| **StableSR** (IJCV2024) | 24.66 | 0.7003 | 0.3101 | 0.6169 | 65.24 | 5.924 | 0.4302 |
| **FedSR+StableSR** | 23.77 | 0.6832 | 0.3502 | 0.6683 | 70.27 | 5.094 | 0.5186 |
| **△StableSR** | -0.89 | -0.0171 | +0.0401 | **+0.0514** | **+5.03** | **-0.830** | **+0.0884** |
| **SUPIR** (CVPR2024) | 23.64 | 0.6603 | 0.3511 | 0.6316 | 61.34 | 6.299 | 0.4952 |
| **FedSR+SUPIR** | 22.25 | 0.6173 | 0.3727 | 0.6807 | 65.57 | 5.685 | 0.5625 |
| **△SUPIR** | -1.39 | -0.043 | +0.0216 | **+0.0491** | **+4.23** | **-0.614** | **+0.0673** |
| **SeeSR** (CVPR2024) | 25.14 | 0.7182 | 0.2996 | 0.6697 | 69.86 | 5.419 | 0.5437 |
| **FedSR+SeeSR** | 24.73 | 0.7258 | 0.2982 | 0.6612 | 70.93 | 5.280 | 0.5591 |
| **△SeeSR** | -0.41 | +0.0076 | -0.0014 | -0.0085 | **+1.07** | **-0.139** | **+0.0154** |
| **PASD** (ECCV2024) | 26.53 | 0.7597 | 0.2783 | 0.5030 | 60.61 | 6.018 | 0.3894 |
| **FedSR+PASD** | 26.15 | 0.7596 | 0.2751 | 0.5191 | 64.78 | 5.744 | 0.4199 |
| **△PASD** | -0.38 | -0.0001 | -0.0032 | **+0.0161** | **+4.17** | **-0.274** | **+0.0305** |
| **DiffBIR** (Arxiv2023) | 24.83 | 0.6473 | 0.3678 | 0.7017 | 69.22 | 5.812 | 0.5584 |
| **FedSR+DiffBIR** | 23.97 | 0.6405 | 0.3667 | **0.7090** | **72.83** | **5.068** | **0.5812** |
| **△DiffBIR** | -0.86 | -0.0068 | -0.0011 | **+0.0073** | **+3.61** | **-0.744** | **+0.0228** |

being blurry). Since FedSR does not modify the original parameters of the models, these erroneous textures are inadvertently amplified. However, for models such as PASD and SUPIR, we successfully preserve the natural fur texture while simultaneously enhancing the quality of other fine details. In summary, with better baselines, FedSR is able to output much more realistic details. And one may also regard our FedSR as a detector to verify the true ISR ability of baseline models.

## 5.3 ABLATION STUDY

In this section, we present our ablation results on StableSR to show the effectiveness of FedSR. First, we validate the effectiveness of the AEM in ISR tasks. Compared to the default settings, removing the AEM results in poorer no-reference metrics (see Row 3 of Table 4), while adding it leads to a noticeable improvement in no-reference metrics. For more visual results, please refer to the Appendix E. Next, to validate the effectiveness of the HEM, we removed this module and it results in worse no-reference metrics compared to the default settings (see Row 2 of Table 4). In contrast, simply adding the HEM leads to a noticeable improvement in no-reference metrics.

Table 4: Ablation studies of FedSR on DRealSR and RealSR benchmarks.

| Variants | | DRealSR/RealSR | | | | | | |
|---|---|---|---|---|---|---|---|---|
| AEM | HEM | PSNR↑ | SSIM↓ | LPIPS↓ | CLIPIQA↑ | MUSIQ↑ | NIQE↓ | MANIQA↑ |
| | | 27.93 / 24.66 | 0.7442 / 0.7003 | 0.3280 / 0.3101 | 0.6272 / 0.6169 | 58.28 / 65.24 | 6.475 / 5.924 | 0.3890 / 0.4302 |
| ✓ | | 27.36 / 24.27 | 0.7322 / 0.6890 | 0.3635 / 0.3389 | **0.6760 / 0.6760** | 65.23 / 69.59 | 5.707 / 5.241 | 0.4681 / 0.5087 |
| | ✓ | 26.91 / 23.85 | 0.7230 / 0.6859 | 0.3655 / 0.3351 | 0.6749 / 0.6522 | 64.56 / 68.55 | 5.768 / 5.386 | 0.4447 / 0.4738 |
| ✓ | ✓ | 26.68 / 23.77 | 0.7206 / 0.6832 | 0.3903 / 0.3502 | 0.6690 / 0.6683 | **67.27 / 70.27** | **5.373 / 5.094** | **0.4810 / 0.5186** |

## 6 CONCLUSION

In this work, we propose a *generic and training-free* framework FedSR for enhancing diffusion-based ISR models from a frequency perspective. To achieve this, we first propose a novel channel selection mechanism for enhancing the amplitude information (AEM). Also, we develop a new semantic-aware high-pass filtering algorithm that adaptively determines the thresholds by feature inputs (HEM). As shown in the extensive experimental evaluation, we demonstrate the effectiveness of the FedSR as a plug-in for most diffusion-based ISR models. Additionally, our analysis of the degradation on the frequency domain may also inspire other ISR models, e.g. GAN-based ISR models (see Appendix C.2). We also hope our work will draw more attention from the community toward a broader view of addressing low-level vision tasks like ISR from a frequency perspective.

ETHICAL STATEMENT

Although our proposed method does not strictly fall under generative AI, it can serve as a plug-and-play framework integrated into diffusion-based ISR algorithms developed by Wang et al. (2023c). As diffusion models evolve toward aligning with human preferences, concerns regarding their potential misuse and malicious purposes (such as generative discrimination or inappropriate content) become increasingly prominent. Regarding other potential societal consequences of our work, none of which we feel must be specifically highlighted here.

REPRODUCIBILITY STATEMENT

We provide our implementation details, including the main algorithm and parameters, which can be found in Section 5 and Appendix D. Additionally, our source code is available in the supplementary materials. This information provides the necessary resources for reproducing our results.

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

# A IMAGE PROCESS IN FREQUENCY DOMAIN

## A.1 STUDIES ON IMAGE FREQUENCY DOMAIN ANALYSIS

**Applications of frequency components.** In recent years, frequency-domain information has widely applied in various computer vision tasks, with many studies exploring the impact of different components on image quality through various frequency decomposition methods (Yang & Soatto, 2020; Yu et al., 2022; Wang et al., 2023a; Si et al., 2023; Huang et al., 2024; Cai et al., 2021; Mao et al., 2023a). (1) Decomposition based on composition: Frequency-domain information can be divided into amplitude and phase spectra. FDA (Yang & Soatto, 2020) reduces the distribution discrepancy between the source and target domains by swapping their amplitude spectra. FSDGN (Yu et al., 2022) addresses the dehazing problem by investigating the correlation between amplitude and phase spectra in the frequency domain under foggy degradation. (2) Decomposition based on distance with the frequency center: The frequency domain can also be divided into high-frequency and low-frequency components. FreeU (Si et al., 2023) suppresses low-frequency features in the frequency domain to prevent Stable Diffusion from generating overly smooth images. FouriScale (Huang et al., 2024) applies low-pass filtering in the frequency domain to alleviate repetitive patterns and structural distortions in the generation of high-resolution images by pre-trained diffusion models. (3) Decomposition based on properties: The frequency domain can be separated into real and imaginary components. DeepRFT (Mao et al., 2023a) applies ReLU networks to the real and imaginary parts of the frequency domain separately to achieve effective image deblurring.

Table 5: Classification and Comparison of Frequency-Domain-Based Super-Resolution Methods.

| Domain | Method | Amplitude and Phase Separate | High- and Low-Frequency Separate | Frequency Loss | Training-Free |
|--------|--------|:---:|:---:|:---:|:---:|
| ISR | FSN | × | ✓ | × | × |
| | FDC | × | ✓ | ✓ | × |
| | ARFFT | × | × | ✓ | × |
| | FADN | × | ✓ | ✓ | × |
| | CRAFT | × | ✓ | × | × |
| VSR | DFVSR | × | ✓ | ✓ | × |
| | FTVSR | × | ✓ | × | × |
| | VideoGigaGAN | × | ✓ | × | × |
| | MFPI | × | × | × | × |
| FSR | SFMNet | ✓ | × | ✓ | × |
| ISR | Ours | ✓ | ✓ | × | ✓ |

**Other methods of frequency-based super-resolution.** The main paper summarizes existing methods that apply frequency transform to super-resolution tasks. Table 5 integrates and categorizes frequency-based super-resolution methods from multiple perspectives. It can be observed that other methods fail to consider degradation systematically. In contrast, our training-free method addresses degradation by modulating degraded amplitude and high-frequency components.

## A.2 IMAGE MODELING IN THE FREQUENCY DOMAIN

To better understand the semantic information represented by various frequency-domain components, we perform a visual modeling of them, shown in Figure 7. First, the image is transformed into the frequency domain, and then three types of decomposition are applied: (1) based on composition: amplitude and phase components; (2) based on distance: high- and low-frequency components; and (3) based on properties: real and imaginary components. Afterward, these components are transformed back into the spatial domain directly. The experimental results demonstrate that the amplitude and phase components, as well as the high- and low-frequency components, can convey the semantic information of the image. Specifically, the amplitude component primarily reflects the style characteristics of the image, such as color and contrast, while the phase component reveals the contour information. The low-frequency component captures the overall structure of the image, whereas the high-frequency component highlights the edges and texture details.

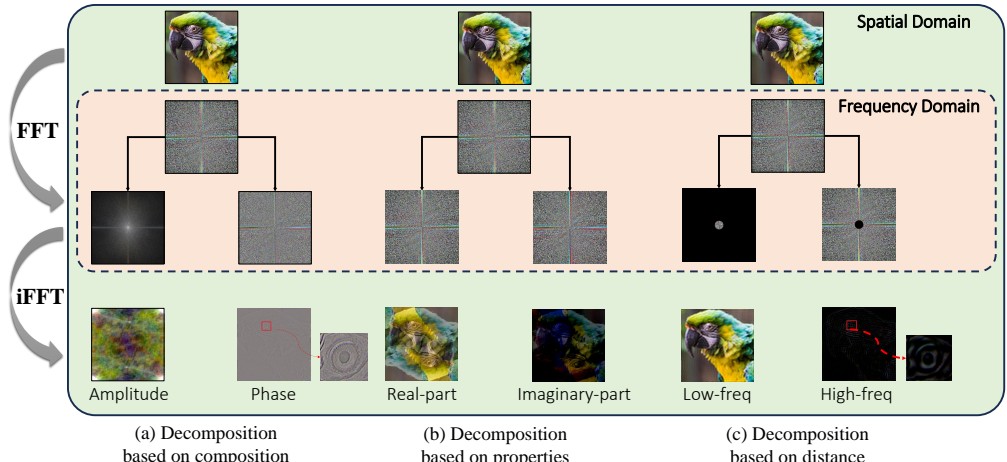

(a) Decomposition
based on composition

(b) Decomposition
based on properties

(c) Decomposition
based on distance

Figure 7: Image modeling methods in the frequency domain. We present the results of three decomposition approaches: (a) decomposition based on composition, (b) decomposition based on properties, and (c) decomposition based on distance. Compared to (b), the method (a) and (c) offer better separation of the intrinsic properties of the image.

## B  QUANTITATIVE RESULTS OF THE IMPACT OF ISR DEGRADATION

In the main paper, the impact of the ISR degradation process on various components is visualized. Detailed quantitative results from testing on RealSR (Cai et al., 2019) are presented in Table 6. The first two rows of Table 6 show the results of replacing the amplitude and phase components of HR images with their corresponding LR counterparts. Following the replacement of the amplitude component, the resulting image quality metrics are poor, indicating that the information loss is primarily concentrated in the amplitude component. The last two rows of Table 6 display the outcomes of replacing the high- and low-frequency components of HR images with their LR counterparts. After the replacement of the high-frequency component, the image quality metrics also remained low, further confirming that the information loss is primarily concentrated in the high-frequency component.

Table 6: Quantitative results of the impact of ISR degradation on amplitude and phase components, high- and low-frequency components.

| Method | CLIPIQA↑ | MUSIQ↑ | NIQE↓ | MANIQA↑ |
|---|---|---|---|---|
| $\mathcal{A}_{LR}+\mathcal{P}_{HR}$ | 0.2320 | 25.44 | 6.849 | 0.1906 |
| $\mathcal{A}_{HR}+\mathcal{P}_{LR}$ | 0.3060 | 28.78 | 6.410 | 0.2373 |
| $\mathcal{H}_{LR}+\mathcal{L}_{HR}$ | 0.2731 | 25.54 | 9.99 | 0.2447 |
| $\mathcal{H}_{HR}+\mathcal{L}_{LR}$ | 0.4447 | 56.98 | 6.035 | 0.3227 |

To further validate our argument regarding the impact of the ISR degradation, we performed the degradation process on phase and low-frequency components that have a minimal influence on the ISR task. Figure 9 demonstrates that the degradation of the phase component directly results in the loss of image structural information, and the degradation of the low-frequency leads to the disappearance of color information, rendering the issue no longer within the scope of ISR research.

## C  COMPARE WITH GAN-BASED METHODS

### C.1  QUANTITATIVE AND QUALITATIVE COMPARISONS ON GAN-BASED MODELS

**GAN-based ISR Methods.**  Based on the results in Table 1, 2, and 3, we further demonstrate the effectiveness of applying our method to DiffBIR and compare its superiority with GAN-based approaches, including BSRGAN (Zhang et al., 2021), Real-ESRGAN (Wang et al., 2021), FeMaSR

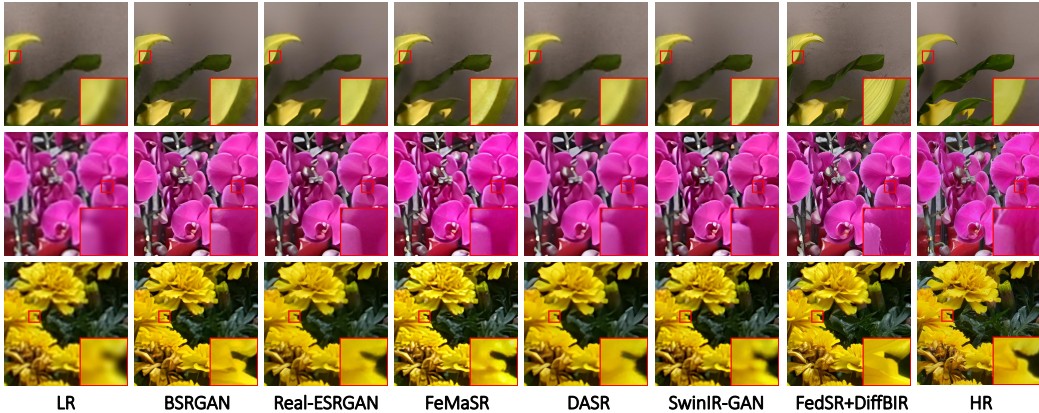

LR    BSRGAN    Real-ESRGAN    FeMaSR    DASR    SwinIR-GAN    FedSR+DiffBIR    HR

Figure 8: Qualitative comparisons of GAN-based methods and our FedSR applied to DiffBIR (Lin et al., 2023) on real-world examples.

(Chen et al., 2022), DASR (Liang et al., 2022), and SwinIR-GAN (Liang et al., 2021). We conduct the tests using publicly available codes and models from the comparison methods.

**Quantitative Comparisons.** Compared to GAN-based methods, our approach demonstrates superior performance in no-reference metrics across three datasets, as shown in Table 7. We also find that GAN-based methods generally perform better in PSNR and SSIM scores. This is because diffusion models generate more realistic details that may not perfectly match the ground truth (GT), thus leading to lower full-reference metrics compared to GAN-based methods.

**Qualitative Comparisons.** To validate the effectiveness of our method, we present a comparison between our approach and GAN-based methods in Figure 8. Our method has a significant advantage in detail generation. Specifically, the DiffBIR (Lin et al., 2023) model combined with our FedSR can produce sharp contours and realistic details, as shown in the second-to-last column of Figure 8, whereas other methods tend to generate blurred results.

### C.2 Discussion on the FedSR Application Effectiveness of GAN

According to Li et al. (2023c), in the training process of most GAN models, the discriminator tends to overemphasize high-frequency components, which weakens the generator's ability to fit low-frequency components. As a result, while GANs can generate sharper images compared to Diffusion models, these images often exhibit unnatural details or artifacts. To explore the model's generalizability, we applied the proposed method to the classic GAN-based ISR model BSRGAN (Zhang et al., 2021), by modulating its high-frequency and amplitude components to enhance the generated results and achieve fine control over each component. Specifically, we introduce the AEM module into the RRDB backbone network to adjust the amplitude components. We also incorporate the HEM module into the residual connections to reduce the impact of high-frequency components. Figure 11 presents the visual results, and the quantitative comparisons are detailed in Table 8.

## D Implementation Details

### D.1 Discussions of Our Amplitude Modulation

**The relationship between low amplitude and high-frequency components.** Although physics indicates that, under the same energy (power), the amplitude of high-frequency waves is usually smaller than that of low-frequency waves, there is currently no evidence in frequency domain analysis of image processing to suggest a one-to-one correspondence between low-amplitude components and high-frequency details, especially in the feature maps of black-box deep learning models. For

Table 7: Quantitative comparison with other GAN-based methods on both synthetic and real-world benchmarks. The **bold** and underline represent the best and second-best performance, respectively.

| DIV2K-Val | | | | | | | |
|---|---|---|---|---|---|---|---|
| **Methods** | **PSNR↑** | **SSIM↑** | **LPIPS↓** | **CLIP-IQA↑** | **MUSIQ↑** | **NIQE↓** | **MANIQA↑** |
| BSRGAN | **24.57** | 0.6232 | 0.3354 | 0.5255 | 61.23 | 4.751 | 0.3561 |
| Real-ESRGAN | 24.29 | **0.6328** | **0.3115** | 0.5283 | 61.11 | **4.674** | 0.3823 |
| FeMaSR | 23.05 | 0.5816 | 0.3125 | 0.5997 | 60.82 | 4.746 | 0.3457 |
| DASR | 24.46 | 0.6267 | 0.3542 | 0.5036 | 55.20 | 5.033 | 0.3186 |
| SwinIR-GAN | 23.92 | 0.6235 | 0.3159 | 0.5340 | 60.22 | 4.706 | 0.3656 |
| DiffBIR+Ours | 22.38 | 0.5222 | 0.4236 | **0.7382** | **73.04** | 4.729 | **0.5838** |
| **DrealSR** | | | | | | | |
| **Methods** | **PSNR↑** | **SSIM↑** | **LPIPS↓** | **CLIP-IQA↑** | **MUSIQ↑** | **NIQE↓** | **MANIQA↑** |
| BSRGAN | 28.68 | 0.8021 | 0.2885 | 0.5104 | 57.25 | 6.518 | 0.3407 |
| Real-ESRGAN | 28.61 | 0.8044 | 0.2848 | 0.4525 | 54.26 | 6.701 | 0.3422 |
| FeMaSR | 26.87 | 0.7557 | 0.3179 | 0.5534 | 53.32 | **5.775** | 0.3121 |
| DASR | **29.74** | **0.8257** | 0.3143 | 0.3807 | 42.43 | 7.522 | 0.2822 |
| SwinIR-GAN | 28.46 | 0.8036 | **0.2801** | 0.4389 | 52.65 | 6.388 | 0.3265 |
| DiffBIR+Ours | 24.53 | 0.6014 | 0.5024 | **0.7167** | **71.90** | 5.833 | **0.5902** |
| **RealSR** | | | | | | | |
| **Methods** | **PSNR↑** | **SSIM↑** | **LPIPS↓** | **CLIP-IQA↑** | **MUSIQ↑** | **NIQE↓** | **MANIQA↑** |
| BSRGAN | 26.37 | 0.7643 | 0.2652 | 0.5105 | 63.19 | 5.690 | 0.3800 |
| Real-ESRGAN | 25.65 | 0.7592 | 0.2720 | 0.4491 | 60.49 | 5.910 | 0.3769 |
| FeMaSR | 25.06 | 0.7342 | 0.2896 | 0.5450 | 59.20 | 5.807 | 0.3648 |
| DASR | **27.01** | 0.7702 | 0.3047 | 0.3135 | 40.95 | 6.682 | 0.2459 |
| SwinIR-GAN | 26.30 | **0.7719** | **0.2479** | 0.4367 | 58.83 | 5.800 | 0.3455 |
| DiffBIR+Ours | 23.97 | 0.6405 | 0.3667 | **0.7090** | **72.83** | **5.068** | **0.5812** |

Table 8: Quantitative results of BSRGAN (Zhang et al., 2021) method on the RealSR with FedSR.

| Method | PSNR↑ | SSIM↑ | LPIPS↓ | CLIPIQA↑ | MUSIQ↑ | NIQE↓ | MANIQA↑ |
|---|---|---|---|---|---|---|---|
| **BSRGAN** | 26.37 | 0.7643 | 0.2652 | 0.5105 | 63.19 | 5.690 | 0.3800 |
| **BSRGAN+ours** | 25.33 | 0.7541 | 0.2711 | 0.5327 | 64.66 | 5.522 | 0.4141 |
| **△ BSRGAN** | -1.04 | -0.0102 | +0.0059 | **+0.0222** | **+1.47** | **-0.168** | **+0.0341** |

instance, in images containing abundant details, textures, and sharp edges, the amplitude of high-frequency components may be large, such as in a dense forest with lush leaves; likewise, the amplitude of high-frequency components increases in the presence of high-frequency noise. In contrast, low-frequency components may have relatively small amplitudes in patterns primarily composed of high-frequency information, such as fine lines or repetitive textures. Therefore, our research attempts to enhance both amplitude and high-frequency components. Our supplementary experiments in Appendix D.2 also indicate that enhancing low-amplitude channels in the backbone is not equivalent to enhancing high-frequency components. This further illustrates the differences between low-amplitude components and high-frequency details.

**The illustrations of reweighting of the AEM** Due to the obscurity of DNN's features, our motivation for optimizing ISR primarily stems from the visible aspects of image space. However, insights derived from image space do not fully apply to the feature space due to the differences between the two. Although the effect of applying amplitude reweighting in AEM is not obvious in the image space (see Figure 10), this operation can effectively enhance the quality of ISR reconstruction in the feature space. This method of reducing amplitude values seems contrary to the conventional approach of increasing values to enhance results, and it is even somewhat counterintuitive. However, Row 2 and 3 in Table 9 indicate that increasing amplitude values actually leads to a deterioration in performance metrics. To explain this phenomenon, we first investigate the denoising process in diffusion models. In the previous paragraph, we detail the differences between low-amplitude

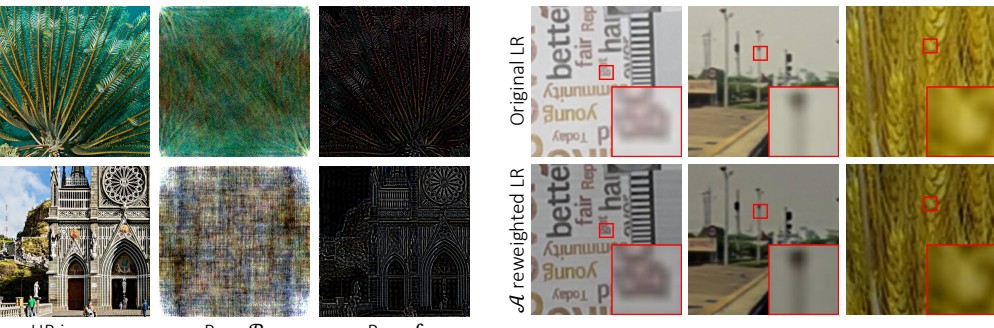

Figure 9: Visualization of degradation in phase and low-frequency components. Phase and low-frequency component degradation is denoted as poor $\mathcal{P}$, and poor $\mathcal{L}$.

Figure 10: Visual comparison between original images and after amplitude $\mathcal{A}$ reweighting on real-world LR images. It shows that the effect is not obvious in image space.

and high-frequency components. Since noise typically belongs to high-frequency components, we cannot establish a direct connection between optimizing denoising and reducing amplitude values. Regrettably, to our knowledge, there is currently no relevant literature evidence suggesting that modulating feature amplitudes to smaller values can improve image quality. Considering the internal structure of U-Net, we hypothesize that the attention modules within U-Net may prefer lower amplitude channel information. Therefore, further reducing the already information-rich low-amplitude channels may assist U-Net in better understanding and representing features, thereby enhancing super-resolution quality. Please note that the preference of deep networks for feature space exceeds the scope of this study, and we will explore this in greater depth in future work. Meanwhile, we encourage researchers in the community to provide more reasonable explanations.

## D.2 DISCUSSION ON THE MODULE CONFIGURATION

To demonstrate the effectiveness of our module configuration, we conduct supplementary experiments on AEM and HEM by replacing the positions of their effects. Specifically, we apply the AEM to the skip features and observe that its performance metrics are inferior to those obtained with the default settings for skip features (see Row 1, 2 of Table 9). Similarly, when applying HEM to the backbone features, the generated results were very poor (see Row 3, 5 of Table 9). This indicates that enhancing high-frequency components in the backbone does not equate to enhancing lower-amplitude components, further confirming the differences between high-frequency and low-amplitude components. Furthermore, we note that FreeU (Si et al., 2023) also includes frequency-related operations. To validate the effectiveness of our adaptive masking for high-frequency components, we replace HEM with the operations of skip features in FreeU, resulting in a significant decrease in no-reference metrics, specifically MUSIQ and MANIQA (see Row 5, 6 of Table 9).

Table 9: Supplementary experiments of the AEM and HEM on DRealSR and RealSR benchmarks.

| Strategy | | DRealSR/RealSR | | | | | | |
|---|---|---|---|---|---|---|---|---|
| Module | Place | PSNR↑ | SSIM↑ | LPIPS↓ | CLIP-IQA↑ | MUSIQ↑ | NIQE↓ | MANIQA↑ |
| AEM | Skip | 27.32 / 24.09 | 0.7236 / 0.6805 | 0.3508 / 0.3275 | 0.6683 / 0.6554 | 62.51 / 67.88 | 6.275 / 5.642 | 0.4154 / 0.4504 |
| AEM | Backbone (A↑) | 28.12 / 24.80 | 0.7456 / 0.6886 | 0.3709 / 0.3404 | 0.4806 / 0.4703 | 45.91 / 55.26 | 6.561 / 6.284 | 0.2944 / 0.3218 |
| AEM | Backbone (A↓) | 27.36 / 24.27 | 0.7322 / 0.6890 | 0.3635 / 0.3389 | **0.6760 / 0.6760** | **65.23 / 69.59** | **5.707 / 5.241** | **0.4681 / 0.5087** |
| HEM | Backbone | 18.67 / 15.98 | 0.4093 / 0.2961 | 0.7266 / 0.7455 | 0.2011 / 0.1907 | 33.92 / 35.64 | 9.410 / 9.612 | 0.2840 / 0.2691 |
| HEM | Skip (FreeU) | 27.46 / 24.44 | 0.7241 / 0.6867 | 0.3520 / 0.3196 | 0.6732 / **0.6563** | 61.10 / 66.29 | **5.613 / 5.294** | 0.3972 / 0.4245 |
| HEM | Skip | 26.91 / 23.85 | 0.7230 / 0.6859 | 0.3655 / 0.3351 | **0.6749** / 0.6522 | **64.56 / 68.55** | 5.768 / 5.386 | **0.4447 / 0.4738** |

## D.3 THE ALGORITHM AND PARAMTERS

As stated in the main paper, the AEM and the HEM are two key modules embedded in the skip connections of the U-Net within the Diffusion model. Each module contains its respective enhance-

---

**Algorithm 1** FedSR Algorithm

---

1: **for** each $t \in [1, \text{Sampling Steps}]$ **do**
2:   Initialize the backbone features $x_{bone}$ and the skip features $\boldsymbol{x}_{skip}$ in the skip connection;
3:   $\boldsymbol{f}_{bone} = \mathcal{F}(\boldsymbol{x}_{bone})$, $\boldsymbol{f}_{\text{skip}} = \mathcal{F}(\boldsymbol{x}_{\text{skip}})$
4:   // (1) Amplitude Enhancement Module
5:   $\mathcal{A}(\boldsymbol{x}_{\text{bone}}), \mathcal{P}(\boldsymbol{x}_{\text{bone}}) = \text{FFTSplit}(\boldsymbol{f}_{bone})$;
6:   // a) Channel Split;
7:   Split $\mathcal{A}(\boldsymbol{x}_{\text{bone}})$ by channel, then obtain $\boldsymbol{S}_A = \{\mathcal{A}(\boldsymbol{x}_{\text{bone}})_i\}_{i=1}^C$;
8:   // b) Average & Order;
9:   **for** each $i \in [1, n]$ **do**
10:     Average amplitude value $a_i = \frac{1}{HW} \sum_{h=1}^H \sum_{w=1}^W \mathcal{A}(\boldsymbol{x}_{\text{bone}})_i^{(h,w)}$;
11:   **end for**
12:   $\text{Order}(\boldsymbol{S}_A)$ by $a_i$
13:   // c) Channel Selection;
14:   $\boldsymbol{S}_S = \{\mathcal{A}(\boldsymbol{x}_{\text{bone}})_i | a_i \leq a_{\min} + P_s \times (a_{\max} - a_{\min})\}$;
15:   // d) Amplitude Modulation;
16:   **if** $\mathcal{A}(\boldsymbol{x}_{\text{bone}})_i \in \boldsymbol{S}_S$ **then**
17:     $\boldsymbol{M}_{\text{bone}} = 1 - P_a \cdot (\overline{\mathcal{A}} - \overline{\mathcal{A}}_{\min})/(\overline{\mathcal{A}}_{\max} - \overline{\mathcal{A}}_{\min}), \mathcal{A}(\boldsymbol{x}_{\text{bone}})_i' = \mathcal{A}(\boldsymbol{x}_{\text{bone}})_i \odot \boldsymbol{M}_{\text{bone}}$;
18:   **end if**
19:   $\boldsymbol{f}_{\text{bone}}' = \text{FFTCombine}(\mathcal{A}(\boldsymbol{x}_{\text{bone}})', \mathcal{P}(\boldsymbol{x}_{\text{bone}})), \boldsymbol{x}_{\text{bone}}' = \mathcal{G}(\boldsymbol{f}_{\text{bone}}')$
20:   // (2) High-frequency Enhancement Module
21:   **for** $r \in [0, S/2]$ **do**
22:     $\boldsymbol{M}_{\text{skip}}(r) = 1 + (r > r_{\text{thresh}}) \cdot [(S - S_{\min})/(S_{\max} - S_{\min}) + 0.5] \cdot P_b/2$
23:   **end for**
24:   $\boldsymbol{f}_{\text{skip}}' = \boldsymbol{f}_{\text{skip}} \odot \boldsymbol{M}_{\text{skip}}, \boldsymbol{x}_{skip}' = \mathcal{G}(\boldsymbol{f}_{skip}')$;
25: **end for**

---

ment parameters, as detailed in Table 10. The detailed algorithmic process for these two modules is shown in Algorithm D.2.

Table 10: The parameters and their definitions for the AEM and HEM, which are set within five state-of-the-art diffusion-based ISR models.

| Module | Parameter | Definition | StableSR | DiffBIR | SUPIR | SeeSR | PASD |
|--------|-----------|------------|----------|---------|-------|-------|------|
| AEM | $P_a$ | The linearization param in Eq. (6) | 0.3 | 0.3 | 0.05 | 0.3 | 0.3 |
|     | $P_s$ | The selection threshold of Figure 6 | 0.3 | 0.3 | 0.3 | 0.3 | 0.3 |
| HEM | $P_{b_1}$ | The scaling factor in Eq. (8) | 0.9 | 0.9 | 0.3 | 0.1 | 0.1 |
|     | $P_{b_2}$ | The scaling factor in Eq. (8) | 0.2 | 0.2 | 0.2 | 0.4 | 0.2 |

## D.4 DISCUSSION ON THE METRICS

We show the DISTS metrics on the RealSR dataset (see Table 11). In the literature, the trade-off between fidelity and visual quality remains a long-standing challenge in the field of SR, and there is currently no definitive optimal evaluation metric. As noted by (Blau & Michaeli, 2018), this trade-off implies that solely optimizing distortion metrics may not only be ineffective but could also degrade visual quality. Meanwhile, we find recent Diffusion-based SR methods tends to emphasizing more on perceptual metrics such as MUSIQ and CLIP-IQA. (Wang et al., 2023c; Yu et al., 2024). Notably, our fluctuations on metrics like PNSR/SSIM are deems acceptable, much lower than the gap between SOTA diffusion-based methods themselves (e.g. SUPIR and StableSR differ by 0.1109 in SSIM, while DiffBIR and StableSR differ by 2.03 points in PSNR).

## D.5 DISCUSSION ON THE TIMESTEP

Our FedSR is a highly flexible framework which can be adapted to specific timesteps. We conduct a preliminary experiment and observe that our FedSR demonstrates a greater impact during the early

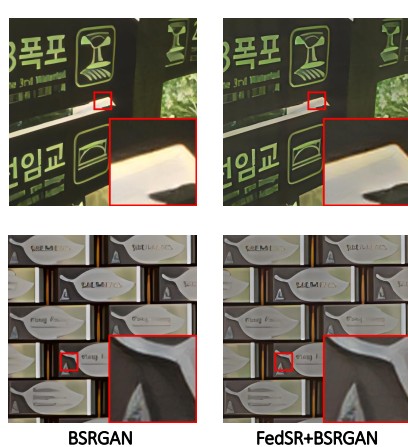

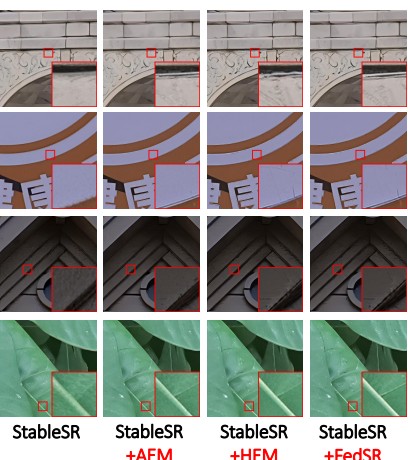

Figure 11: Visual comparisons of BSR-GAN. After applying our method, BSRGAN (Zhang et al., 2021) presents the results with more natural image details and contrast.

Figure 12: Visual effects of AEM and HEM. The results show that AEM primarily enhances the overall appearance, while HEM improves the clarity of details.

Table 11: Quantitative results of DISTS metrics on RealSR dataset.

| Matrics | StableSR | DiffBIR | SeeSR | PASD | SUPIR |
|---|---|---|---|---|---|
| Baseline | 0.2202 | 0.2401 | 0.2227 | 0.1989 | 0.2494 |
| FedSR+ | 0.2422 | 0.2562 | 0.2294 | 0.2029 | 0.2632 |

denoising stages (see last row of Table 12). Additionally, incorporating FedSR in segments proves less effective than applying it as a whole.

Table 12: Quantitative results on specific timesteps on the RealSR dataset.

| Matrics | PSNR | SSIM | LIPIPS | MUSIQ | CLIP-IQA | NIQE | MANIQA |
|---|---|---|---|---|---|---|---|
| StableSR | 24.66 | 0.7003 | 0.3101 | 65.24 | 0.6169 | 5.924 | 0.4302 |
| StableSR+FedSR(totally 1-200) | 23.77 | 0.6832 | 0.3502 | 70.27 | 0.6683 | 5.094 | 0.5186 |
| StableSR+FedSR (1-100) | 24.04 | 0.6860 | 0.3302 | 68.62 | 0.6489 | 5.343 | 0.4681 |
| StableSR+FedSR (101-200) | 24.35 | 0.6966 | 0.3272 | 68.65 | 0.6628 | 5.511 | 0.5031 |

## D.6 COMPLEXITY ANALYSIS

In this section, we evaluate the complexity of the FedSR method using StableSR and DiffBIR as examples. We list the parameters and FLOPs of the denoising models in each framework below, which demonstrate almost the same statistics after integrating FedSR.

Table 13: Parameters and FLOPS of denoising models before and after integrating FedSR.

| Matrics | StableSR | FedSR+StableSR | DiffBIR | FedSR+DiffBIR |
|---|---|---|---|---|
| Param (M) | 918.93 | 918.93 | 1666.75 | 1666.75 |
| FLOPs (G) | 375.55 | 375.59 | 61.45 | 61.49 |

## E  ADDITIONAL VISUAL RESULT

In this section, we present additional experimental results. Figure 12 illustrates the visual effects of AEM and HEM when applied individually and in combination. The results show that AEM primarily enhances the overall image appearance, such as contrast, while HEM mainly improves the clarity of high-frequency details.

## F  LIMITATIONS AND FUTURE WORK

Although our proposed FedSR achieves significant results, there are still some limitations. Similar to other ISR studies on natural scenes, this work focuses only on existing natural image datasets and synthetic datasets for ISR tasks. Applying ISR on a larger scale to AI-generated datasets remains an interesting avenue for further exploration. Additionally, we only employ a training-free implementation, without delving into model training and fine-tuning. In future work, we will explore how to leverage the network's preference for frequency domain components to fine-tune the model architecture, thereby further enhancing ISR quality.

