# OpenReview forum: "FedSR: Frequency-Aware Enhancement for Diffusion-based Image Super-Resolution"
_ICLR.cc/2025/Conference — ICLR 2025 Conference Withdrawn Submission_

### Official Review · Reviewer_sedK · 2024-11-01

**Soundness:** 2
**Presentation:** 2
**Contribution:** 2
**Rating:** 5
**Confidence:** 5

**Summary:**

This paper tackles the problem of image super-resolution (ISR) by focusing on the frequency domain. While traditional diffusion-based ISR methods primarily address spatial domain issues, this paper introduces a new approach to enhance ISR performance by emphasizing image amplitude information and high-frequency details, achieving notable performance.

**Strengths:**

The paper is well-structured and clearly explains the methodology.

The proposed method brings a considerable improvement in certain metrics.

**Weaknesses:**

1. Lack of Novelty in Motivation: The motivation presented in Figure 1 is not novel and has appeared in numerous low-level vision papers, including but not limited to:

- Exploring Fourier Prior for Single Image Rain Removal.

- Deep Fourier-based Exposure Correction Network with Spatial-Frequency Interaction

These papers are not discussed in the paper either. The insights in the contribution section (page 107) seem incremental.

2. Although FedSR+ enhances existing SR methods on metrics like CLIP-IQA, it significantly lowers fidelity on PSNR and SSIM, which is contrary to the paper’s motivation. For instance, in Table 1, using the proposed method reduces StableSR’s PSNR by 0.67dB, which is quite unacceptable.

3. Lack of Complexity Analysis: The paper does not discuss the computational complexity introduced by FedSR, such as parameters and FLOPs. If the method significantly increases computational overhead while offering only marginal improvements, it would have limited practical deployment value.

4. Potential Unrealistic Details:
- Figure 4 shows images without ground truth (GT).
- It appears that applying the proposed method may introduce “unrealistic” texture details, thus improving metrics like CLIP-IQA, which are less sensitive to fidelity, while decreasing PSNR, which is fidelity-sensitive.
5. Need for User Study:   It is crucial to conduct a user study to validate that the enhanced details provided by the proposed method are “real” and consistent with the GT. Emphasizing fidelity is extremely important in SR tasks; without it, the task may as well be a generation task.
6. There are several existing papers on real-SR in the frequency domain, but the paper lacks discussion and comparison in this regard, including but not limited to:
-  Image Super-resolution Via Latent Diffusion: A Sampling-space Mixture Of Experts And Frequency-augmented Decoder Approach.
-  Wavelet-based Fourier Information Interaction with Frequency Diffusion Adjustment for Underwater Image Restoration.
-  Waving Goodbye to Low-Res: A Diffusion-Wavelet Approach for Image Super-Resolution.

**Questions:**

As shown in Weaknesses

---

### Official Review · Reviewer_9YAT · 2024-11-01

**Soundness:** 3
**Presentation:** 3
**Contribution:** 2
**Rating:** 5
**Confidence:** 5

**Summary:**

The authors propose a frequency-aware enhancement framework, FedSR, to enhance the performance of the existing diffusion-based SISR methods. The main motivation is that the amplitude information and high-frequency information in the diffusion model have a greater impact on the quality of super-resolved images. To this end, the authors introduce two modules, i.e., a channel-aware amplitude enhancement module and a high-frequency enhancement module with adaptive masking, for diffusion-based SISR methods. To verify the effectiveness of the proposed method, the authors conducted quantitative and qualitative analysis experiments on several datasets.

**Strengths:**

1. To investigate the influence of various components in the image frequency domain on the diffusion-based SISR method, the author performed an analytical experiment, providing an interesting perspective.
2. The proposed Amplitude Enhancement Module (AEM) and High-Frequency Enhancement Module (HEM) can be simultaneously integrated into current diffusion-based SISR models in a training-free manner.
3. Experiments show that the proposed FedSR achieves better quantitative results on multiple no-reference metrics.

**Weaknesses:**

1. The authors argue that the amplitude information and high-frequency information in the image are crucial for addressing the image degradation problem through experimental analysis. However, it is questionable whether these components always play a pivotal role at different time step T in the denoising process, because the function of the diffusion model is different at different time steps.
2. Although the proposed method achieves good results in the no-reference metrics, there are negative optimization results in the visual comparison shown in Figure 4. For example: (1) within the results of StableSR+ours, DiffBIR+ours, and SeeSR+Ours concerning image1, the animal hair appears more artificial and unrealistic; (2) within the results of SeeSR+Ours concerning image1, the background area where the plants are located is incorrectly generated; (3) within the results of SeeSR+Ours concerning image4, the loss of high-frequency details pertaining to the plants in the lower left corner contrary to the viewpoint proposed in the paper.
3. How are the values of P_b and r_thresh determined in Equation 8? The authors are encouraged to conduct additional experimental analyses to enhance comprehension surrounding these parameters.

**Questions:**

Please refer to the weaknesses.

---

### Official Review · Reviewer_qSQn · 2024-11-03

**Soundness:** 2
**Presentation:** 3
**Contribution:** 2
**Rating:** 3
**Confidence:** 5

**Summary:**

The authors propose a plug-and-play approach to improve the quality of diffusion-based super-resolution. By selectively leveraging intermediate network features advantageous for reconstruction and amplifying high-frequency components within these features, the proposed method aims to enhance generative quality. The paper presents comprehensive evaluations of the model's performance across various established methods, showcasing its effectiveness.

**Strengths:**

1. The proposed method demonstrates substantial improvements on non-reference metrics. Extensive experiments further confirm its generalizability across a range of scenarios.
2. Incorporating effective skip connections is a reasonable and beneficial modification for the super-resolution task, where fidelity is crucial. This adaptation aligns well with the goal of preserving high-quality details in reconstructed images.

**Weaknesses:**

1. The observations presented in Figure 1, such as the significance of high-frequency information for image recovery, are well-known in the field and thus cannot be considered novel contributions to this study.
2. As shown in Tables 1, 2, and 3, the proposed method exhibits no improvement—and in some cases, a decline—in performance on reference-based metrics. Although pixel-aware metrics like PSNR and SSIM may not fully align with human perception, perceptual metrics such as LPIPS and DISTS should be included to better evaluate the fidelity of the results. Relying solely on non-reference metrics is insufficient for assessing recovery quality, as these metrics can be influenced by artifacts and noisy textures, which also applies to the results in Table 4.
3. Frequency-based approaches for super-resolution have become quite prevalent, as exemplified by methods like [1][2]. The lack of novelty in the frequency-focused aspect of the proposed method limits its contribution to the field.
4. In Figure 4, particularly the first, third, and fourth rows of the squirrel example in the first column, the textures of the squirrel pelts have transformed into pseudo-textures after applying the proposed method, raising further doubts about the reliability of the improved non-reference metrics.

[1] Image Super-resolution via Latent Diffusion: A Sampling-space Mixture of Experts and Frequency-Augmented Decoder Approach

[2] Frequency-Domain Refinement with Multiscale Diffusion for Super Resolution

**Questions:**

1. Could you clarify how you concluded that a lower average amplitude in the channel results in clearer generated images based on the experiment depicted in Figure 3? The visualizations seem to indicate that as the channel average amplitude decreases, the information in the images becomes more blurred, which appears to contradict the stated conclusion.
2. Could you specify the meaning of 'the radius' in Lines 343–344?
3. The model incorporates several hyperparameters; are these adjusted manually, or is there a systematic approach to optimizing these hyperparameters for different super-resolution backbones?

---

### Official Review · Reviewer_j9md · 2024-11-05

**Soundness:** 2
**Presentation:** 3
**Contribution:** 2
**Rating:** 5
**Confidence:** 5

**Summary:**

This paper introduces a novel frequency-aware enhancement framework called FedSR for diffusion-based image super-resolution (ISR) methods. FedSR uses two key components - the Amplitude Enhancement Module (AEM) and the High Frequency Enhancement Module (HEM) to selectively enhance and adaptively filter the amplitude information and high-frequency details of the image, respectively, significantly improving the restoration of image details. degree without the need for additional training or fine-tuning.

**Strengths:**

1. Improving diffusion model super-resolution via frequency feature
2. A significant advantage of this method is that it requires no additional training or fine-tuning

**Weaknesses:**

1. There are many papers on improving image recovery from the frequency domain perspective, and frequency domain feature selection mechanisms have also been proposed.
2. The PASD results in Table 2 and Table 3 are quite different from the PASD original paper. I did not carefully check the experimental results of other methods.
3. There are already many methods for super-resolution based on diffusion models. The frequency domain algorithm proposed in this paper is just a combination of past algorithms and diffusion models, and there is no special innovation.
4. Like this method, there are already many methods that use frequency enhancement to generate details and downstream tasks. For example, FreeU: Free Lunch in Diffusion U-Net

**Questions:**

Regarding the experimental results issue
And what is the most unique originality of this paper

---

### Note · Authors · 2025-01-12

I have read and agree with the venue's withdrawal policy on behalf of myself and my co-authors.